# αO-Conotoxin GeXIVA Inhibits the Growth of Breast Cancer Cells via Interaction with α9 Nicotine Acetylcholine Receptors

**DOI:** 10.3390/md18040195

**Published:** 2020-04-07

**Authors:** Zhihua Sun, Jiaolin Bao, Manqi Zhangsun, Shuai Dong, Dongting Zhangsun, Sulan Luo

**Affiliations:** 1Medical School, Guangxi University, Nanning 530004, China; 2Key Laboratory of Tropical Biological Resources of Ministry of Education, Key Laboratory for Marine Drugs of Haikou, School of Life and Pharmaceutical Sciences, Hainan University, Haikou 570228, China; zhihuasun918@163.com (Z.S.); fish1012@hotmail.com (J.B.); zhangsunmanqi@163.com (M.Z.); dongshuai_1024@163.com (S.D.)

**Keywords:** α9-nicotinic acetylcholine receptors (nAChRs), breast cancer cells, αO-conotoxin GeXIVA, apoptosis, anti-proliferation, targeted therapy

## Abstract

The α9-containing nicotinic acetylcholine receptor (nAChR) is increasingly emerging as a new tumor target owing to its high expression specificity in breast cancer. αO-Conotoxin GeXIVA is a potent antagonist of α9α10 nAChR. Nevertheless, the anti-tumor effect of GeXIVA on breast cancer cells remains unclear. Cell Counting Kit-8 assay was used to study the cell viability of breast cancer MDA-MD-157 cells and human normal breast epithelial cells, which were exposed to different doses of GeXIVA. Flow cytometry was adopted to detect the cell cycle arrest and apoptosis of GeXIVA in breast cancer cells. Migration ability was analyzed by wound healing assay. Western blot (WB), quantitative real-time PCR (QRT-PCR) and flow cytometry were used to determine expression of α9-nAChR. Stable MDA-MB-157 breast cancer cell line, with the α9-nAChR subunit knocked out (KO), was established using the CRISPR/Cas9 technique. GeXIVA was able to significantly inhibit the proliferation and promote apoptosis of breast cancer MDA-MB-157 cells. Furthermore, the proliferation of breast cancer MDA-MB-157 cells was inhibited by GeXIVA, which caused cell cycle arrest through downregulating α9-nAChR. GeXIVA could suppress MDA-MB-157 cell migration as well. This demonstrates that GeXIVA induced a downregulation of α9-nAChR expression, and the growth of MDA-MB-157 α9-nAChR KO cell line was inhibited as well, due to α9-nAChR deletion. GeXIVA inhibits the growth of breast cancer cell MDA-MB-157 cells *in vitro* and may occur in a mechanism abolishing α9-nAChR.

## 1. Introduction

Cancer is the second leading cause of death worldwide, and it was estimated to account for 9.6 million deaths in 2018 [1]. According to the statistics, breast cancer is the second most common carcinoma in the world after lung cancer, and is also the highest-incidence cancer among women. In a global context, breast cancer remains the leading cause of cancer incidence and mortality, with 2.1 million newly diagnosed cases and 630 thousand deaths in 2018 [1,2]. Recurrence and metastasis are the major cause of these deaths [2]. Additionally, genetic and reproductive risk factors play important roles in susceptibility to breast cancer [3]. Conventional therapeutic strategies usually provide limited specificity, resulting in severe side effects and toxicity to normal organisms. In contrast, targeted cancer therapy could improve the therapeutic potential of anti-tumor agents and reduce adverse side effects [4].

Nicotinic acetylcholine receptors (nAChRs) belong to ligand-gated ion channels, which are composed of transmembrane subunits that share a common evolutionary origin [5]. The different nAChRs subunits can form homo- or hetero-pentamers and enclose a central ion channel. nAChRs are expressed in the cell membrane of all mammalian cells, including cancer cells [6,7,8]. nAChRs not only mediate normal physiological responses, such as inflammation and pain, but also participate in the regulation of Alzheimer disease, Parkinson’s disease, schizophrenia and cancers etc [8,9,10]. 

α7-nAChR and α9-nAChR are the major nAChRs in breast cancer cells [11,12,13]. Increasing evidence suggests that activation of α7-nAChR leads to activation of the ERK/MAPK and JAK2/PI3K signaling cascades in breast cancer cells [7,14]. In addition, it was reported that α9-nAChR was highly correlated with breast cancer [7,11,15], and stimulation of the α9-nAChR led to breast cancer growth [9]. Further research suggests that a low dose of garcinol (1 µM) from the edible fruit *Garcinia indica* could inhibit nicotine-induced breast cancer cell proliferation through the downregulation of α9-nAChR and cyclin D3 expression [7]. Luteolin and quercetin also could inhibit the ability of proliferation by downregulating the expression of α9-nAChRs on the cell surface of human breast cancer cells [15]. Tea polyphenol(-)-epigallocatechin-3-gallate has been found to inhibit nicotine-and estrogen-induced α9-nicotinic acetylcholine receptor upregulation in human breast cancer cells and delay the development of breast cancer cells in vivo [13]. These results implied that α9-containing nAChRs detected in human breast cancer cells could be used as a new therapeutic molecular target for cancer treatment. 

As antagonists to nAChRs, α-conotoxins (α-Ctxs) are used to decipher the pharmacological functions of these receptors, and some of them also have therapeutic potential [16,17]. αO-conotoxin GeXIVA is a potent antagonist of α9α10 nAChRs, which was discovered in *Conus generalis*, native to the South China Sea [16]. This peptide is composed of 28 amino acids including four Cys residues that can form three different disulfide bond connection isomers, i.e., GeXIVA[1,2], GeXIVA[1,3], and GeXIVA[1,4]. They were tested for their potency against both rat and human α9α10 nAChRs, as described previously. Among them, GeXIVA[1,2] was the most potent antagonist of α 9 α 10 nAChR (IC 50 = 4.61 nmol at rat α 9 α 10 nAChR) with a high specificity, as described previously [16,18,19]. It displayed potent alleviation of neuropathic pain in rat model in vivo [20]. Recent reports revealed that both GeXIVA, targeting α9α10 nAChRs, and α-conotoxin TxID, targeting α3β4 nAChRs, contributed to the inhibition of cancer cell proliferation [21,22]. Compared with currently widely used macromolecular antibodies, small peptide toxins have a unique advantage to be developed as new drugs to treat various diseases, since small peptides can overcome the limitations of poor tumor penetration and cellular uptake of antibody when introduced in vivo [23,24]. Antagonists of nAChRs may be novel potential drugs for tumor-targeting therapy and diagnosis because of their specific binding activities and other natural properties. 

Our previous research found that α9-nAChR was highly expressed in breast cancer MDA-MB-157 cells. In this study, we aimed to investigate the potential of GeXIVA for treatment of α9-nAChR overexpressed breast cancer cell in vitro. Firstly, the stability of GeXIVA in cell culture medium was investigated. Then, the influences of GeXIVA on growth and apoptosis of MDA-MB-157 cells were examined. In addition, we explored the potential carcinogenic effects of the α9-nAChR subunit through a stable α9 nAChR-knockout (KO) cell line of MDA-MB-157 cells.

## 2. Results

### 2.1. A Detection of the Stability of GeXIVA in Cell Culture Medium by RP-UPLC

The stability of GeXIVA was tested in cell culture medium (Figure 1). RP-UPLC (Reverse phase high-performance liquid chromatography) was used to determine the extent of the degradation of the peptides, then, a time-changing trend diagram of GeXIVA in culture media with different concentrations of serum was drawn to judge the stability of the GeXIVA in cell cultures. After 24 h incubation in different concentrations of serum (H_2_0 Control, 0% FBS, 5% FBS, 10% FBS), the remaining amounts of GeXIVA were ~93%, ~73%, ~66% and ~54% respectively. The results showed that the content of GeXIVA decreased while the serum concentration increased, and as time went on the trend continued. To assure maximum efficacy of drugs, we selected DMEM medium without serum in the process of GeXIVA-treated cells.

### 2.2. GeXIVA Affected the Growth of Breast Cancer Cells MDA-MB-157

We compared the effects of GeXIVA on human epithelial cancer cell line MDA-MB-157 and human normal mammary gland epithelial cell line HS578BST (Figure 2). Microscopic examination revealed that cell density and cell morphology were changed significantly after 24 h treatment of GeXIVA at different concentrations, and cancer cells were more sensitive to GeXIVA than normal cells. In the MDA-MB-157 breast cancer cell line, cell proliferation was remarkably decreased after treatment of GeXIVA in a concentration- and time-dependent manner (Figure 2A). In MDA-MB-157 cells, the IC_50_ of 24 h for GeXIVA was approximately 78.31 µM. These results indicated that GeXIVA could effectively inhibit the proliferation of breast cancer cells MDA-MB-157, but had no toxic effects on normal cells HS578BST.

### 2.3. GeXIVA Induced Apoptosis in MDA-MB-157 Cells

Apoptosis is a major cause of cancer cell growth inhibition, and previous studies have confirmed that α9-nAChR affects cell proliferation in MDA-MB-157 breast cancer cells. Herein, two-color flow cytometry with Annexin V-FITC and Propidium iodide (PI) labeling showed necrosis to be the predominant mode of cell death in MDA-MB-157 cells treated with various concentrations of GeXIVA (11.25, 22.5, 45 and 90 µM) for 24 h. The significant difference is shown in the representative scatter plots of cells treated by a series of concentrations of GeXIVA (Figure 3A–E). The percentage of early/late apoptosis cells was summarized in Figure 3F. In control group, the proportion of early and late apoptotic cells was 0.73%. After 24 h treatment with 11.25–90 µM GeXIVA, the ratio of early and late apoptotic cells was significantly increased by up to 27.05%. These results showed that GeXIVA inhibits the growth of MDA-MB-157, probably by inducing cell apoptosis.

### 2.4. GeXIVA Induced Cell Cycle Arrest in MDA-MB-157 Cells

To elucidate whether GeXIVA treatment induces mitotic inhibition during cell division, we performed cell cycle analysis. Flow cytometry analysis demonstrated that the 24 h incubation of MDA-MB-157 cells with GeXIVA significantly increased the number of cells in the S phase of cell cycle, while the number of cells in the G0/G1 phase was significantly decreased (Figure 4A,B). Regarding G2/M phase, the cells number was significantly decreased when the MDA-MB-157 cells were treated with GeXIVA at the concentration of 90 µM. The results indicated that GeXIVA induced a cell cycle arrest in S phase in MDA-MB-157 cells. 

### 2.5. GeXIVA Decreased the Breast Cancer Cells Abilities of Migration

To test the potential changes in some carcinoma-associated characteristics of the MDA-MB-157 cells after GeXIVA treatment, cell migration ability was measured. The results indicated that cells migrated more slowly to close the scratched wounds after treatment with GeXIVA for 24 h compared with control group (Figure 5). Then, the effects of different concentration on the migration distance of MDA-MB-157 cells were analyzed statistically. This indicated that GeXIVA had inhibition effects on the migration of MDA-MB-157 cells; following the increasing concentrations, cells migration was retarded (Table 1).

### 2.6. GeXIVA Affected Breast Cancer Cell Line MDA-MB-157 Proliferation through the Inhibition of α9-nAChR-mediated Signals

To test the effect of α9-nAChR while GeXIVA inhibiting breast cancer cells, a stable cell line with *α9-nAChR* gene knock-out was established. The α9-nAChR CRISPR/Cas9 KO plasmid and HDR plasmid were transfected into MDA-MB-157 cells and the transfected cells were selected under the suppression of puromycin. The positive cells were confirmed by detection of the red fluorescent protein (RFP) via fluorescent microscopy (Appendix A). Real-time PCR showed that transfection of MDA-MB-157 cells with α9-nAChR CRISPR/Cas9 KO and HDR plasmid led to more than a 10-fold decrease in α9-nAChR mRNA expression, while the expression of α3 and β4 nAChR subunits mRNA(as control) had no differences (Figure 6A). Flow cytometry showed that knockout of the *α9-nAChR* gene greatly reduced the expression of functional α9-nAChRs on the cell surface (Appendix A and Figure 6B). We also applied Western blotting analysis to confirm the expression of α9-nAChR protein was significantly silenced in knocked out cell lines (MDA-MB-157 α9-nAChR KO) (Figure 6C). 

The influence of the α9-nAChR knockout on viability was determined in the breast cancer cell line by commercial Cell Counting Kit-8 (CCK-8) kit. As shown on Figure 7A, MDA-MB-157 WT and MDA-MB-157 α9-nAChR KO cell lines illustrated similar dynamic curves of viability. CCK-8 assay was used to determine the anti-proliferative effect of GeXIVA in both MDA-MB-157 WT and MDA-MB-157 α9-nAChR KO cell lines. GeXIVA displayed cytotoxicity in a concentration-dependent manner in the MDA-MB-157 cell line. However, knocked out *α9-nAChR* gene completely reversed the anti-proliferative effect of GeXIVA (Figure 7B). In addition, GeXIVA produced a survival rate of 1.2% to 18% in MDA-MB-157 wild type cells at high concentrations (90 and 180 µM), while high concentrations of GeXIVA had little effect on the proliferation of MDA-MB-157 α9-nAChR KO cells. Taken together, we can conclude that the anti-proliferative activity of GeXIVA on MDA-MB-157 cells is a result of the peptide interaction with α9-nAChRs. 

To further test the possible influence of GeXIVA on the expression of α9-nAChR, breast cancer cell line MDA-MB-157 was treated with GeXIVA for 24 h. Cells were harvested and subjected to total RNA extraction. qRT-PCR assays for the α9-nAChR showed significantly reduced transcriptions after cells treated with GeXIVA; however, β4-nAChR was almost unchanged before and after the treatment (Figure 8A). The expression of α9-nAChR was determined by indirect fluorescence staining and flow cytometry analysis, the results revealed a significant decrease in the level of α9-nAChR, after 24 h exposure of cells to GeXIVA at concentrations of 22.5 µM and higher (Figure 8B).

## 3. Discussion

The α9-nAChR subunit, initially discovered in hair cells of the inner ear, can form a homo-pentamer or assemble into a hetero-pentamer with α10 or α5-nAChR subunits [25,26]. Including α9-nAChR, almost all of nAChRs are expressed not only in neuronal systems, but also in numerous non-neuronal tissues cells such as skin, pancreas and lung, suggesting that nAChRs may have roles in other biological processes in addition to synaptic transmission [8]. It is reported that α9-nAChR is involved in promoting cancer cell proliferation, angiogenesis, cancer metastasis and apoptosis suppression during carcinogenesis in response to tumor microenvironments [7,11,13,27,28]. In a previous study, we determined the expression profile of the α9-nAChR subunit in human breast cancer cell lines and found that α9-nAChR was expressed differently in breast cancer cells. This suggests that α9-nAChR may become an effective target for breast cancer treatment [29]. 

GeXIVA, a *Conus* peptide, contains 28 amino acids and four Cys residues which have three disulfide bonds [16]. The original study of GeXIVA on α9α10 nAChR expressed in *Xenopus* oocytes revealed that GeXIVA by itself is able to block the α9α10 nAChR current of ACh-evoked currents completely at a 100 nM concentration, and this block was rapidly reversible [16]. GeXIVA is a potent and selective antagonist of α9α10 nAChR subtype [16,30]. In vivo, it also displayed potent alleviation of neuropathic pain in several rat models [18,20]. In vitro, GeXIVA showed an antitumor effect [22]. Therefore, it is worth further investigating GeXIVA as a potential therapeutic agent against breast cancer, etc.

Previous research has indicated that GeXIVA is unstable in serum [30]. In this study, we decided to evaluate the stability of GeXIVA in cell culture medium, and the ability of the GeXIVA to control the growth of MDA-MB-157 cells. The remaining amount of GeXIVA reached ~80% in serum-free DMEM medium after 48 h, so, in the next steps, the cells were co-incubated with GeXIVA in serum-free DMEM medium to assure maximum efficacy. Prolonged 24 h incubation of MDA-MB-157 cells with GeXIVA resulted in the pronounced concentration-dependent inhibition of the cell growth with IC_50_ of ~78 µM. Moreover, GeXIVA had an obvious inhibitory effect on breast cancer cell line MDA-MB-157 compared with normal breast epithelia cell line HS578BST in vitro. This means that the same doses of GeXIVA could kill more cancer cells than normal cells. In other words, GeXIVA is less toxic to normal cells than cancer cells. Moreover, apoptosis analysis demonstrated that the apoptosis rates in the drug treatment group were significantly higher than in the control group. This result indicated that GeXIVA inhibit the growth of MDA-MB-157, probably by inducing cell apoptosis. We also demonstrated that inhibition of α9-nAChR by GeXIVA could inhibit MDA-MB-157 cancer cell proliferation through the induction of G0/G1 arrest. Metastasis is a major obstacle in clinical cancer therapy [31]. Previous studies have indicated that nicotine can enhance the migratory abilities of cancer cells [31,32,33]. However, some studies reported the antagonists of nAChRs could reduce the progression of cancer cells [34]. Here, we found that GeXIVA could abolish the abilities of cancer migration.

Previous studies have shown that natural polyphenol compounds can reduce nicotine-mediated carcinogenic effects by inhibiting the expression of α9-nAChR [13,15,35,36]. It is reported that α9-containing nAChRs may play an important role in breast cancer [11,13], but we knew little about the antitumor effect of GeXIVA previously. To determine whether the anti-proliferative effect of GeXIVA is associated with α9-nAChRs, we blocked expression of this receptor by CRISPR/Cas9 knockdown technique. The anti-proliferative effects were profoundly abolished by GeXIVA treatment in *α9-nAChR* knockdown MDA-MB-157 cells compared with wild-type cells. Thus, we can conclude that the GeXIVA antiproliferative effect on MDA-MB-157 cells is a result of the protein interaction of α9-nAChR with GeXIVA. Therefore, the GeXIVA inhibitory effects might be through downregulating of the protein and gene expression of α9-nAChR. More experiments are needed to gain insight into the underlying molecular mechanisms. 

## 4. Materials and Methods

### 4.1. Cell Culture

Human breast cancer cell line MDA-MB-157 and human normal mammary gland epithelial cell line HS578BST were purchased from the Kunming Institute of Zoology (Yunnan, China). To make the complete growth medium, all the cells were maintained in Dulbecco’s modified eagle medium (DMEM), supplemented with fetal bovine serum (FBS) to a final concentration of 10%, 100 U/mL penicillin and 100 mg/mL streptomycin at 37 °C in an atmosphere of 5% CO_2_.

### 4.2. Cell Culture Medium Stability

Medium stability assay was performed in cell culture media with different concentrations of serum for GeXIVA [1,2]. With setting experimental group (DMEM + GeXIVA, 5% FBS + DMEM + GeXIVA, 10% FBS + DMEM + GeXIVA) and comparison group (GeXIVA was incorporated into the aqueous phase), the sample of each group was placed in an incubator at 37 °C with 5% CO_2_ atmosphere and 100% relative humidity for 0, 12, 24 and 48 h. 30 µL triplicate aliquots were taken out at different time points. Each serum aliquot was quenched with 30 µL of 6 M urea and incubated for 10 min at 4 °C. Then, each serum aliquot was treated with 30 µL of 20% trichloroacetic acid (TCA) for another 10 min at 4 °C to precipitate serum proteins. Precipitated serum proteins were then spun down at 14,000 g [19]. Qualitative and quantitative analysis were performed by RP-UPLC, through drawing time-changing trend diagrams of GeXIVA in culture media with different concentrations of serum to judge the stability of the GeXIVA in cell cultures. 

### 4.3. Cell Counting Kit-8 Assay (CCK-8)

To study effects of GeXIVA on cell growth, 5 × 10^3^ cells were seeded in 96-well culture plates. After 24 h, the cells were incubated in a complete medium at 37 °C for adhesion. Thereafter, GeXIVA (dissolved in water) was added to cells at different concentrations (11.25, 22.5, 45, 90 and 180 µM) and grown during 24 h in medium without FBS at 37 °C, 5% CO_2_. The medium was removed and the CCK-8 solution was added to cell wells for 1 h at 37 °C. The absorbance at 450 nm (OD 450) each well was obtained using a microplate reader. Do not add any compounds to cells as control. The blank group was supplemented with culture medium and contained no cells.

Cell viability was measured with a commercial Cell Counting Kit-8. Briefly, 2 × 10^3^ cells were seeded onto 96-well plates and the CCK-8 solution was directly added to the cell wells and incubated at 37 °C for 1 h. The absorbance at 450 nm (OD_450_) for each well was obtained by using a microplate reader with a background control as blank. The cell viability was expressed as the percentage of the untreated control. 

### 4.4. Migration Assay

Wound assay was used to evaluate the migration ability of cells. A total of 1 × 10^5^ cells were seeded in a 12-well plate and grew to confluence followed by scratching the monolayer cells with a 200 μL pipette tip to create a wound. After the medium and floating cells were removed, the cells were rinsed with phosphate-buffered saline three times. The cells were treated with different concentrations of GeXIVA and incubated for 24 h in medium without FBS at 37 °C, 5% CO_2_. Plates were washed to remove floating cells and debris, then cell migration images were taken at 0, 24 and 48 hours after drug treatment. Cells without addition of any compounds were used as a control. Each testing group contained at least three independent wells.

### 4.5. Flow Cytometry

Apoptosis assay was detected with an Annexin V-FITC/PI Apoptosis Detection Kit (KeyGENBioTECH, Nanjing, China) and flow cytometry analysis. A total of 2 × 10^5^ cells were seeded onto 6-well plates. After a 24 h incubation for adhesion, the cells were treated with GeXIVA for 24 h. Cells were collected by trypsinization and dual stained with Annexin V-FITC and PI for 15 min at room temperature in dark. For each sample, data from approximately 1 × 10^4^ cells were recorded in list mode on logarithmic scales. Apoptosis and necrosis were analyzed by quadrant statistics on PI-negative, Annexin V-positive cells and cells positive for both, respectively.

Cells were seeded in 6-well culture plates and incubated with different concentrations of GeXIVA for 24 h. Then the cells were detached from the wells by Trypsin Solution (Without EDTA), washed with PBS containing 3% bovine serum albumin (BSA), and fixed in pre-cooling 70% ethanol for 4 h. After fixation, the cells were washed twice by 3% BAS-PBS, resuspended in 10% RNase-PI staining solution, and analyzed by Guava easyCyte^TM^ flow cytometer.

The expression of α9-nAChR in cells was determined by indirect fluorescence staining and flow cytometry analysis. The treated cells were harvested and washed with PBS twice, fixed with 4% PFA for 20 min and washed twice by centrifugation at 1000 rpm for 5 min each time. Cells were suspended in 0.1% Triton X-100 in 1 × PBS buffer for 10 min and washed twice with PBS by centrifugation at 1300 rpm for 5 min each time. The cells were incubated with 3% BSA in 1 × PBS buffer for 30 min. Primary antibody at an appropriate dilution was added and incubated for 2 hr at room temperature, then washed and incubated with secondary antibody (FITC-conjugated Goat anti-rabbit Ig (G+L)) for 1 hr at room temperature. Finally, the cells were washed twice and re-suspended in 300 μL 1 × PBS and analyzed the expression of α9-nAChRs by flow cytometry. The primary antibody used was as follows: CHRNA9 Rabbit Polychonal Antibody (26025-1-AP, proteintech, Chicago, IL, USA).

### 4.6. Generation of Stable α9-nAChR-KO Cell Lines

MDA-MB-157 cells were plated in a 6-well culture plate at a density of 1.5–2.5 × 10^5^ cells in 3 mL antibiotic-free standard growth medium per well. After 24 h, the cells were co-transfected with α9- nAChR CRISPR/Cas9 Knock out (KO) plasmid (Santa Cruz, sc-402753-KO-2, CA, USA) and α9- nAChR homology-directed repair (HDR) plasmid (Santa Cruz, sc-402753-HDR-2, CA, USA). After incubation for 24 h at 37 °C in a CO_2_ incubator, the plasmid-containing medium was removed and fresh growth medium containing 10% serum was added in the presence of 6 μg/mL of puromycin. Selected cells, for a minimum of 3–5 days, approximately every 2–3 days, were aspirated and replaced with freshly prepared selective medium. This resulted in the establishment of MDA-MB-157 KO cells in which α9-nAChR gene expression was knocked out.

### 4.7. RNA Isolation and Quantitative Real-time PCR

Total RNA of cells was isolated using Trizol (Invitrogen, Carlsbad, CA, USA) according to the manufacturer’s protocol. Equal amounts of RNA (1 µg) from each sample were reverse-transcribed into first-strand cDNA with a High-Capacity cDNA Reverse Transcription Kit (Applied Biosystems, Carlsbad, CA, USA). The expression of the target gene was evaluated by qRT-PCR on Analytikjena qTOWER^3^G using specific primers. Primers used were as follows: GAPDH forward: (CAGCCTCAAGATCATCAGCA) and reverse: (TGTGGTCATGAGTCCTTCCA); α9-nAChR forward: (TGGCACGATGCCTATCTCAC) and reverse: (TGATCAGCCCATCATACCGC), α3 nAChR forward: (AACGTGTCTGACCCAGTCATCAT) and reverse: (AGGGGTTCCATTTCAGCTTGTAG); β4-nAChR forward: (TCACAGCTCATCTCCATCAAGCT) and reverse: (CCTGTTTCAGCCAGACATTGGT). The relative expression level of α9, α3 and β4- nAChR mRNAs were determined by the comparative Ct (2^−△△Ct^) method. 

### 4.8. Protein Extraction and Western Blot Analysis

The cultured cells were washed with cold phosphate buffer saline (PBS) three times and harvested using a cell lysis buffer containing protease inhibitors PMSF (Solarbio Life Sciences, Beijing, CHN). Equal amounts of protein from control and treated cell lysates were separated using a 12.5% sodium dodecyl sulfate polyacrylamide gel electrophoresis (SDS-PAGE) gel under reducing conditions and transferred onto polyvinylidene fluoride (PVDF) membranes (Solarbio Life Sciences, Beijing, China), which were subsequently probed with primary antibodies (AChRα9, Santa Cruz, sc-293282, CA, USA). In all Western blots, membranes were additionally probed with an antibody for GAPDH (Santa Cruz, sc-47724, CA, USA) to ensure equal loading of protein between samples. Horseradish peroxidase-conjugated secondary antibodies (m-IgGκ BP-HRP, Santa Cruz, sc-516102, CA, USA) were used with enhanced chemoluminescence reagent (Biosharp, Guangzhou, China) to visualize the protein bands. Images of the films were captured using the Alpha FluorChem E (ProteinSimple, CA, USA).

### 4.9. Statistical Analysis

Three to five independent repeats were conducted for all experiments. Error bars represent these repeats. Statistical comparisons between groups were performed using ANOVA (Prism GraphPad Software and SPSS 17.0). A Student’s *t*-test was used and a *p*-value < 0.05 was considered significant. Analysis was performed with the SPSS software package (version Version 17.0).

## 5. Conclusions

In this study, the anti-proliferative activity of GeXIVA on breast cancer cells was confirmed *in vitro*, which deserves further study as a potential agent for anti-cancer therapy. Our results demonstrated that α9-nAChR played a key role in breast cancer cells, which was differentially over-expressed in breast cancer cell lines than in normal cells. As an antagonist of α9-nAChR, αO-conotoxin GeXIVA induced cell apoptosis and inhibited the proliferation of breast cancer MDA-MB-157 cells. On the other hand, we found that the proliferation of breast cancer cells was blocked by GeXIVA through S-phase cell cycle arrest. Moreover, GeXIVA induced a downregulation of α9-nAChR expression, and the anti-tumor effect of GeXIVA was abolished by α9-nAChR silencing. These findings provide obvious molecular evidence that GeXIVA is effective drug lead for targeting therapy of breast cancer, which can inhibit the growth of breast cancer cells.

## Figures and Tables

**Figure 1 marinedrugs-18-00195-f001:**
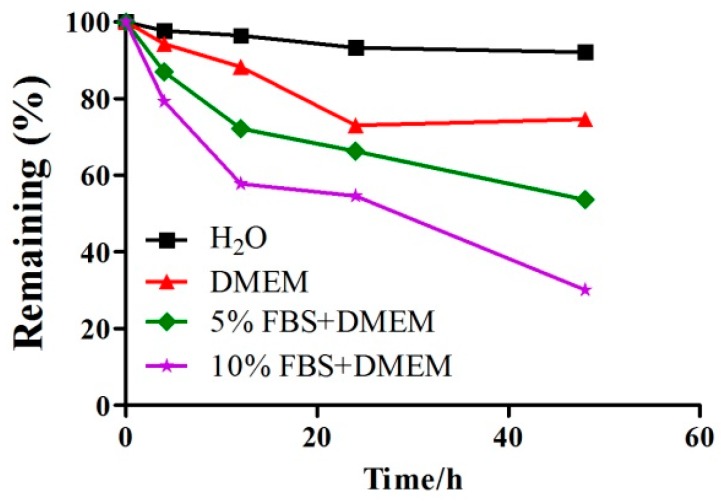
The stability of GeXIVA in cell culture medium. Time course of effects on GeXIVA in cell culture media with different concentration of fetal bovine serum (0%, 5%, 10% FBS), under a 5% CO_2_ atmosphere and 37 ℃ incubation conditions. Each point represents the mean for each time point.

**Figure 2 marinedrugs-18-00195-f002:**
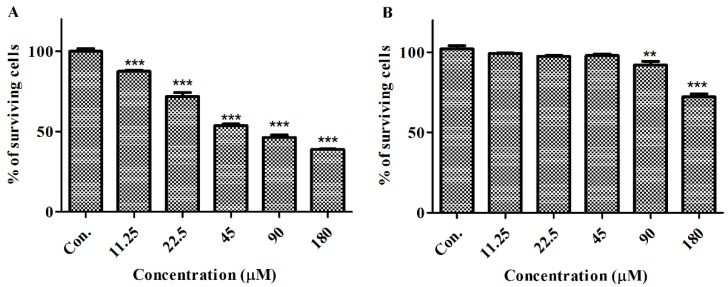
Effects of GeXIVA on the viability of breast cancer cells and normal cells. The cells were treated with various concentrations of GeXIVA for 24 h. Then, the cell viability was determined by CCK-8 assay. Values were expressed as mean ± SD of three independent assays. Statistical analysis was performed with one-way ANOVA and Tukey’s test. ** *p* < 0.01, *** *p* < 0.001 indicating a significant difference between the treatments compared to medium control. (**A**) MDA-MB-157; (**B**) Hs578BST.

**Figure 3 marinedrugs-18-00195-f003:**
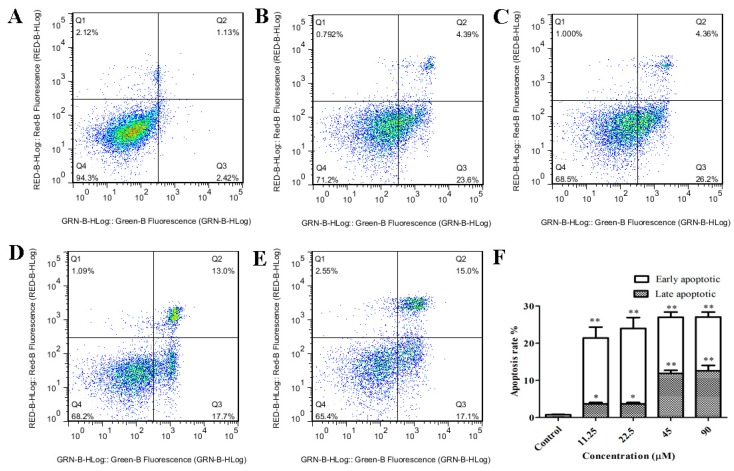
Flow cytometry measurements of apoptosis in MDA-MB-157 cells treated with GeXIVA. Data are presented as dot plots in which the vertical axis represents fluorescence due to PI staining and the horizontal axis represents the fluorescence associated with Annexin V-FITC. The upper left quadrant (Q1) contains necrotic (PI positive) cells, the upper right region (Q2) contains late apoptotic (mixture of PI and Annexin V positive) cells. The lower left region (Q4) contains healthy living (PI and Annexin V negative) cells, and the lower right region (Q3) contains early apoptotic (PI negative and Annexin V positive) cells. Cells were pretreated with 11.25 µM (**B**), 22.5 µM (**C**), 45 µM (**D**), 90 µM (**E**) GeXIVA for 24 h. Then, the cells were washed, harvested, and re-suspended in PBS. The amount of apoptosis cells was measured by flow cytometer. Data were expressed as mean ± SEM of three independent experiments. Significant different was performed by one-way ANOVA. * *p* < 0.05 and ** *p* < 0.01 compared to the control group. **A:** Control. **F:** The inhibition rate was examined by FCM (Flow Cytometry).

**Figure 4 marinedrugs-18-00195-f004:**
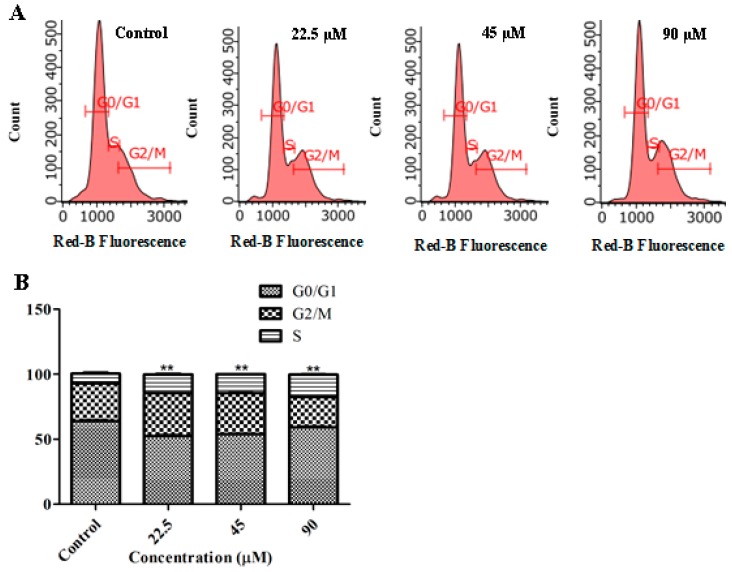
Cell cycle arrest in MDA-MB-157 cells treated with GeXIVA. (**A**) Representative nuclei population distributions of MDA-MB-157 cells before and after incubation with GeXIVA. (**B**) % of cells in each cell cycle phase determined by flow cytometry. The data are presented as % of cells in each cell cycle phase ± SEM, *n* = 3; ** (*p* < 0.01) indicates a significant difference between the control and GeXIVA-treated groups by one-way ANOVA.

**Figure 5 marinedrugs-18-00195-f005:**
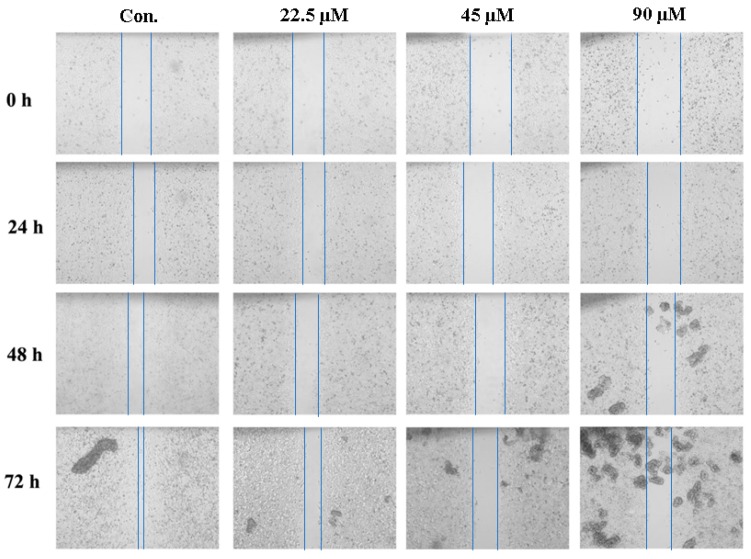
Evaluation of the malignancy of the breast cancer cells after treatment with GeXIVA. Migration assay. MDA-MB-157 cells were pretreated with different concentrations of GeXIVA for 24 h. After treatment, the migration ability of the cells was tested at 24 h, 48 h and 72 h. The blue vertical lines in the images define the open wound areas.

**Figure 6 marinedrugs-18-00195-f006:**
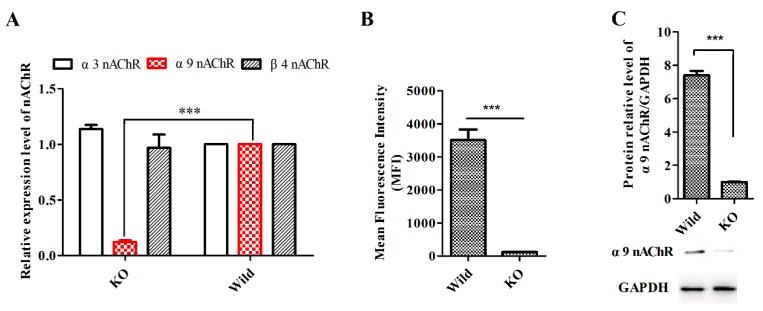
Establishment of a stable α9-nAChR silently expressed MDA-MB-157 breast cancer cell line. **A**. Detection of the mRNA expression of α9-nAChR by qRT-PCR, with the mRNA expression of α3-nAChR and β4-nAChR as control. Data are presented as mean ± SEM, *n* = 3. *** (*p* < 0.001) indicates the significant difference between KO and wild by two-tailed one sample t-test. **B**. Median fluorescence intensities for FITC-labeled antibody against the α9-nAChR for untreated cells (control) and cells with the knocked out α9-nAChR expression by CRISPR/Cas9. Data are presented as mean ± SEM, *n* = 3. *** (*p* < 0.001) indicates the significant difference from control by two-tailed t-test. **C**. Western blot detection of the protein expression of cells after knocked out *CHRNA9* gene treatment. Experiments were repeated three times. *** (*p* < 0.001) indicates the significant difference between KO and wild by one-way ANOVA.

**Figure 7 marinedrugs-18-00195-f007:**
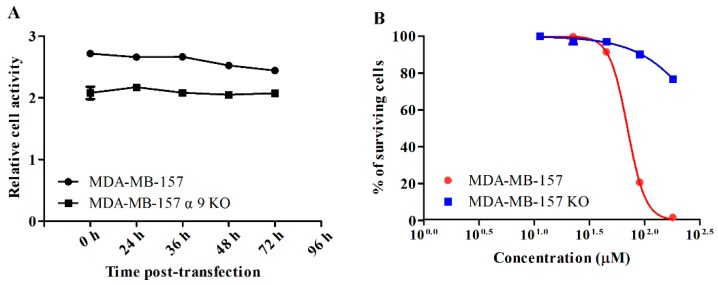
Evaluation of the malignancy of the MDA-MB-157 cells after transfection. (**A**) Cell viability assays. Cells were assayed with Cell Counting Kit-8 kit at various time points after transfection. Each test was repeated three times. Graphical data denote mean ± SD. (**B**) Effects of GeXIVA on cytotoxicity in cells. Breast cancer cells were treated with various concentrations of GeXIVA for 24 h. Then, the anti-proliferation ability of GeXIVA was determined by the CCK-8 assay. Values were expressed as mean ± SD of three independent assays. * *p* < 0.001 for a significant difference in the MDA-MB-157 α9-nAChR KO cell line in comparison to the MDA-MB-157 cell line.

**Figure 8 marinedrugs-18-00195-f008:**
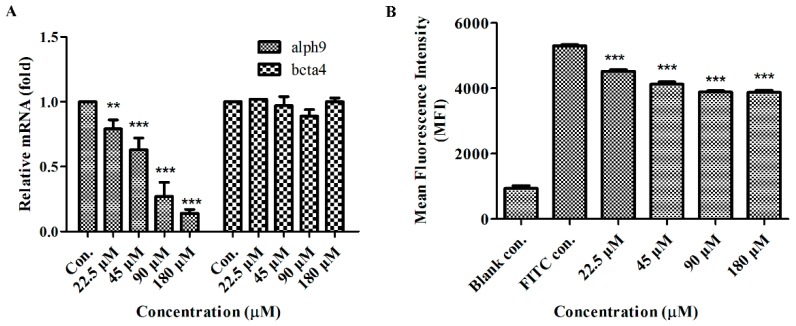
Effects of GeXIVA on the viability of breast cancer cells. **A**, Expression levels of α9 nAChR subunit mRNA were evaluated by real-time PCR analysis. MDA-MB-157 cells were treated with different concentrations of GeXIVA for 24 h. **B**, Flow cytometry analysis of α9-nAChR staining intensity under GeXIVA treatment. Mean staining intensities for α9-nAChR in MDA-MB-157 cells treated with GeXIVA at various concentrations for 24 h compared with the control processed in parallel. Data are expressed as mean ± SD, *n* = 3. ** *p* < 0.01 and *** *p* < 0.001 for significant differences from the control group.

**Table 1 marinedrugs-18-00195-t001:** The effects of different concentration of on the migration distance of MDA-MB-157 cells in vitro.

Migration Distance	Control		22.5 µM		45 µM		90 µM	
	Mean ± SD	%	Mean ± SD	%	Mean ± SD	%	Mean ± SD	%
0 h	0.52 ± 0.00	–	0.59 ± 0.00	–	0.73 ± 0.00	–	0.74 ± 0.00	–
24 h	0.13 ± 0.01^aAB^	24.52	0.17 ± 0.02^aA^	28.41	0.16 ± 0.03^aAB^	21.56	0.13 ± 0.04^aB^	17.19
48 h	0.24 ± 0.01^bA^	45.81	0.17 ± 0.03^aB^	28.41	0.19 ± 0.02^aA^	25.69	0.21 ± 0.06^aA^	28.05
72 h	0.44 ± 0.02^cA^	84.52	0.27 ± 0.02^bB^	45.45	0.28 ± 0.06^aB^	38.07	0.26 ± 0.02^bB^	34.84

Note: The capital letters A, B and C represent the 95% confidence intervals. One-way ANOVA was performed on the same row (same times) and different columns (different concentrations) in the table. The lowercase letters a, b and c represent the 95% confidence intervals. One-way ANOVA was performed on the same columns (same concentrations) and different columns (different times) in the table. Values with same superscript letters in the same line showed no significant differences (*p*
*>* 0.05), those with different letters showed significant or extreme differences (*p* < 0.05).

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
