# Peer review of "αO-Conotoxin GeXIVA Inhibits the Growth of Breast Cancer Cells via Interaction with α9 Nicotine Acetylcholine Receptors"

_marinedrugs, 2020, doi:10.3390/md18040195_

Round 1

Reviewer 1 Report

 The manuscript describes about the growth inhibitory activity of αO-conotoxin GeXIVA against the breast cancer cells, via inhibition and down-regulation of α9 nAchR. As almost all the results seem to be solid and the manuscript was well written, the reviewer can recommend the manuscript for publication in Marine Drugs, after Some comments or insights to the following points should be given.

 The authors described that GeXIVA interact with α9-nAchR selectively, and most of the experiments such as knockdown or protein expression level analysis were conducted only about α9-nAchR. Do you have any information about the expression level of the other subunits of nAchR, or binding/inhibitory activity against them? α7- or α10-nAchR are similar subunits to form ion channels with α9-nAchR, and α5-nAchR is known to be involved in the inhibitory activity of luteolin or quercetin against breast cancer cells.

 Figure 1 shows that GeXIVA is unstable in the presence of serum. Have you ever checked the bioactivity of the degraded peptide fragment? Or, do you have any information about structural requirement of GeXIVA for exhibiting its activity?

Reviewer 2 Report

The article is devoted to the study of the activity of a small peptide discovered from Conus generalis in relation to breast cancer. The authors sequentially studied all important points in order to make a final conclusion about the possibility of using GeXIVA conotoxin as a drug.

It was shown: (1) the peptide stability under experimental conditions, (2) dose-dependent effect on the growth of breast cancer cells (3) apoptosis of cells upon incubation with the peptide and (4) decrease of the ability of breast cancer cells to migrate.

Further, the authors using the knockout of the α9-nAChR receptor gene clearly showed that conotoxin GeXIVA, for which activity as an α9 α10-nAChR antagonist has already been shown, realizes its anticancer activity on this cell line due to interaction with this receptor’s subunit.

The article is a logical and consistent study, the results are clear and not in doubt.
Key points relate to the design and language of the article.

  1. line 85-86 – the remaining amounts of GeXIVA are indicated, while concentration of serum is not indicated (hereinafter it is only in the figure caption)
  2. line 174-175 «Real-time PCR showed that transfection of….» - - the incomplete names of the plasmids used for transfection are indicated; there is no specification of which mRNA was determined in the real-time PCR reaction
  3. line 176 – must be «… the expression  of  α3  and  β4  nAChR  subunits  mRNA  were have  NO  differences». Moreover, only the real-time PCR procedure for α9-nAChR is indicated in the methods, there is no mention of other subunits that were also checked.
  4. To identify plasmids, both α9-nAChR CRISPR / Cas9 and AChRα9 CRISPR / Cas9 are used; uniformity should be given.
  5. line 191 – Is «the transfaction» means «the expression»? Moreover, the predicate is missing in this sentence.
  6. line 197-199 - not quite clear sentence construction
  7. line 200 – «…is a result OF the peptide interaction with α9-nAChRs». Hereinafter, conotoxin should be called a peptide, not a protein.
  8. in many places prepositions are missing and there is no agreement on the tense forms of verbs (lines 301, 331-334, 343 and others)
  9. line 351 – «Primers used were  as  follows…» .instead  «Primers  used  were  as  followed…»
  10. 10 line 297  «Cells withOUT addition of any compounds were used as a control». instead «Cells withOUT addition of any compounds were used as a control»

Reviewer 3 Report

In this study authors provide evidence on the anti-proliferative activity of GeXIVA in vitro, using the MDA-MB-157 cell line as a model of breast cancer. They show that GeXIVA inibits cell proliferation of MDA-MB.157 breast cancer cell lines by promoting apoptosis. Moreover, they show that this effect is related to the down-regulation of a9-nAChR, overexpressed in breast cancer cell lines.

-The study itself is nicely conducted, but more experimental evidences are necessary to be provided to make this study fully convinced. For example, it should be investigated the effect of GeXIVA on nAChR downstream signaling pathways.

-Too much discussion is focused on the results of the authors themselves. Authors should shorten these contents but supplement more discussion on comparing their results with other literatures to further clarify and highlight their novelty.

-More information on GeXIVa must be included in the paper.

-Authors should clarify if they used UPLC or HPLC instrument for the analysis of the GeXIVA stability in cell culture medium (Paraghraph 2.1, lines 81 and 82).

-Authors should reported in the text the concentrations of serum tested (Paraghraph 2.1, line 8.

-The text of Figure 1 is wrong: 0.5% or 5% of FBS.

-Please, check the p value reported in the legend of the figure 2.

- Text check is required.

Reviewer 4 Report

Title of manuscript:

αO-Conotoxin GeXIVA Inhibits the Growth of Breast 2 Cancer Cells via Interaction with α9 Nicotine 3 Acetylcholine Receptors

The authors study the effect of GeXIVA on breast cancer cells. They investigate cell viability, proliferation,cell cycle arrest, apoptosis, migration ability in breast cancer MDA-MB-157 cells and the growth of MDA-MB-157 α9-nAChR KO (α9-26 nAChR deleted). The authors did conclusion that GeXIVA inhibits the growth of breast cancer cell MDA-MB-157 cells in vitro and  may occur in a mechanism through abolishing α9 nAChR.

Several points of this investigation should have additional explanation.

  1. The authors recently test expression of Nicotine Acetylcholine Receptors: α9nAChR expressed in all twenty tested cell lines: Differential Expression of Nicotine Acetylcholine Receptors Associates with Human Breast Cancer and Mediates Antitumor Activity of αO-Conotoxin GeXIVA. (Mar. Drugs 2020, 18, 61; doi:10.3390/md18010061). Why for new investigation the authors choose MDA-MB-157 cell line? In cell lines BT-549 and MDA-MB-231 the expression of α9nAChR was highest.
  2. Authors study cell proliferation in MDA-MB-157 and HS578BST cells, the IC50 of 24 h for GeXIVA were approximately 35.18 mM and ~280 mM, respectively. But according bars of Figure 2 A the % of surviving cells is about 50% at 180 mM.
  3. Figure 3. According data IC50 of 24 h for GeXIVA in MDA-MB-157 was approximately 35.18 mM. Why for apoptosis the used concentrations are 45 and 90 mM? Base of data for cell viability 50% of cells are not survived. Figure 4 5, and 8. The same question about concentrations for cell cycle, migration investigation, and expression level of GeXIV.
  1. Also, what happened with cells (investigation of migration breast cancer cells MDA-MB-157, treated with 90 mM of GeXIVA for 24h in DMEM without FBS) maintained during 72 hours later in DMEM with 10% FBS?
  2. The title of manuscript: αO-Conotoxin GeXIVA Inhibits the Growth of Breast 2 Cancer Cells via Interaction with α9 Nicotine 3 Acetylcholine Receptors. But there not presented experiments with directly interaction of GeXIVA and α9 Nicotine 3 Acetylcholine Receptor. The title should be corrected or additional experiments required to support present title.

Several points related with mistakes in manuscripts.

  1. Throw all manuscript needs change uM to mM (for exsample line 54, 101, 250, Figure 5), change M to mM in Table 1.
  2. Figure 1. Add SD for each points.
  3. Table 1. Please, explain calculation of %: 0.13 is 24.52% at 0.52 (maybe 25%); 0.24 is 45.81% (maybe 46.1)?
  4. Figure 7B. What means concentration 2-1, 20, 21, 22?
  5. Authors should correct information in Materials and Methods.
  6. Line 276. “..the cell were maintained in maintained in Dulbecco's modified eagle medium (DMEM)”
  7. Line 288. “Qualitative and quantitative analysis were performed by UPLC” HPLC?
  8. Line 291. “CCK-8 assay”. Cell Counting Kit-8 assay (CCK-8).
  9. Line 292. “To study effect of GeXIVA on cells growth, 5000 cells were seeded in 96-well culture plates”. 5x103 cells.
  10. Migration assay. How many cells were seeded in 12-well plate?

Round 2

Reviewer 3 Report

The paper can be accepted in this form.

Author Response

Thank you for your valuable and thoughtful comments.

Reviewer 4 Report

From the previous comment 1.

Authors study cell proliferation in MDA-MB-157 and HS578BST cells, the IC50 of 24 h for GeXIVA were approximately 35.18 mM and ~280 mM, respectively. But according bars of Figure 2 A the % of surviving cells is about 50% at 180 µM.

Response: Thank you for pointing this out. The IC50 value got from the curve fitting of test results.

Present comment.

This explanation is not correct. Authors didn’t correct their big mistake. Where the curve for test results? From Figure 2 we can see that IC50 is about 180 µM. From the IC50 the authors choose concentrations for future experiments in this manuscript. If the authors calculate wrong data of IC50 it means that ALL data presented in this manuscript have not correct explanation. The authors should completely rewrite this manuscript.

From the previous minor comments

  1. Throw all manuscript needs change uM to mM (for exsample line 54, 101, 250, Figure 5), change M to µM in Table 1.

Response: Thank you for your question. As you can see, if change uM to mM throw all manuscript, the concentration value will become four-five digits after the decimal point. For example, 0.01125, 0.0225, 0.045 and 0.09 mM. So, uM will become unified concentration units.

Present comment.

Authors dismissively answered to my question. I asked to change uM to µM (not for mM!). Other figures from this manuscript (Figure 2, 3, 4, and 7) have units µM!

Round 3

Reviewer 4 Report

  1. The presented result (IC50 = 78.31 μM) is consistent now with the data in Figure 1.
  2. Other minor points for correction of presented data in figures 7 and 8 in attached file.

Author Response

Thank you for your suggestion.  The data in  figure 7 have been corrected in last revision. Meanwhile, we also have checked all the other parts of this manuscript, and found that correcting the IC50 did not affect the entire manuscript, for example figure 8.